# Technical note: A guide to using three open-source quality control algorithms for rainfall data from personal weather stations

Abbas El Hachem[1,2], Jochen Seidel[1], Tess O'Hara[3], Roberto Villalobos Herrera[4], Aart Overeem[5], Remko Uijlenhoet[6], András Bárdossy[1], and Lotte de Vos[5]

[1]Institute for Modelling Hydraulic and Environmental Systems, University of Stuttgart, D-70569 Stuttgart, Germany
[2]Federal Waterways Engineering and Research Institute (BAW), Karlsruhe, Germany
[3]School of Engineering, Newcastle University, Newcastle upon Tyne NE1 7RU, UK
[4]School of Civil Engineering, Universidad de Costa Rica, Ciudad Universitaria Rodrigo Facio, San José, Costa Rica
[5]Royal Netherlands Meteorological Institute (KNMI), de Bilt, The Netherlands
[6]Department of Water Management, Delft University of Technology, The Netherlands

**Correspondence:** Abbas El Hachem (abbas.el_hachem@baw.de)

**Abstract.** The number of rainfall observations from personal weather stations (PWS) has increased significantly over the past years; however, there are persistent questions about data quality. In this paper, we introduce three quality control algorithms (PWSQC, PWS-pyQC, and GSDR-QC) designed for the quality control of rainfall data. Technical and operational guidelines are provided to help interested users in finding the most appropriate QC to apply for their use case. All three algorithms can be accessed within the OpenSense-sandbox where users can run the code. The result shows that all three algorithms improve PWS data quality when cross-referenced against rain radar. The considered algorithms have different strengths and weaknesses depending on PWS and official data availability, making it inadvisable to recommend one over another without carefully considering the specific setting. The need for further objective quantitative benchmarking of QC algorithms requiring freely available test datasets representing a range of environments, gauge densities, and weather patterns is highlighted.

## 1 Introduction

Precipitation is highly variable in space and time and thus the accurate estimation of precipitation amounts is of fundamental importance for many hydrological purposes (Estévez et al., 2011), especially on smaller scales and high temporal resolutions such as in small catchments and in the field of urban hydrology (Berne et al., 2004; Ochoa-Rodriguez et al., 2015; Cristiano et al., 2017), where typical rain gauge networks are not always sufficiently dense to capture the spatio-temporal variability of precipitation. Weather radar provides rainfall estimates with good spatial coverage, but since radar is an indirect measurement, its data suffer from errors and uncertainties (Fabry, 2015; Rauber and Nesbitt, 2018). One approach to improve precipitation estimates is the use of additional data from so-called opportunistic sensors (OS) such as terrestrial commercial microwave links, personal weather stations (PWS) or satellite microwave links, which are typically more numerous than rain gauges from national weather services. The high number of OS-devices offers a huge potential to better capture the strong spatio-temporal variability of rainfall especially in regions with scarce conventional meteorological observations. This holds true in particular for PWS, where the number of stations has increased considerably over the last years.

Some of the most popular and widely available PWS are simple, low-cost instruments that measure various meteorological parameters, including temperature, wind, and rainfall. The rain gauges of PWS are typically unheated tipping bucket gauges with varying orifice sizes and measurement resolution. Operators of PWS also have the opportunity to share and visualise the data on online platforms. For further details on PWS stations and PWS networks see de Vos et al. (2019) and Fencl et al. (2024).

Since these PWS may be installed by people who do not have access to, or knowledge of, optimal gauge placement, it is expected that many of these stations are not set up and maintained according to professional standards. Furthermore, issues like uncertain or missing meta data, data gaps, variable time steps and biases are frequent and hamper the use of PWS rainfall data for hydrological and meteorological applications (de Vos et al., 2019; O'Hara et al., 2023). Overall, there is a high availability of PWS data, but the expected quality of these data is fairly low.

As with all weather observations, in order to make constructive use of PWS rainfall observations, the application of reliable quality control (QC) is vital. Many national meteorological services and other institutions have operational QC algorithms for their precipitation data, but these are typically not open source and are not tailored for PWS data. This can be because they assume a higher data availability and smaller bias than what is commonly found from PWS devices. In the past years, several QC methods for PWS rainfall data have been proposed, which are typically applied to PWS datasets in different geographical areas or time periods. This lack of overlap in climate, conditions and network density can make it difficult for a reader to compare these methods. Overcoming these limitations and to make data from PWS and other OS sensors available to a broader scientific community are aims of the EU COST Action CA2016 "Opportunistic Precipitation Sensing Network (OpenSense)" (https://opensenseaction.eu), where, for example, data standards (Fencl et al., 2024) and software for processing and quality controlling OS data (https://github.com/OpenSenseAction/OPENSENSE_sandbox) are being developed.

For people new to the field, it can be difficult to appreciate the differences between the available methods and conclude which method best suits their needs. The aim of this paper is to provide a guideline to using three different open-source QC methods designed especially for precipitation data. They can be run in the public sandbox environment of the OpenSense EU COST-Action (https://github.com/OpenSenseAction/OPENSENSE_sandbox). As an example the methods are applied to the same publicly available PWS rainfall dataset from the Amsterdam metropolitan area in the Netherlands. Lastly, by following the open data and open source concept the implementation of the QC algorithms is reproducible. The OpenSense COST-Action strives to promote FAIR principles in research, which are increasingly adopted and required by publishers, funding agencies, and academic institutions (Boeckhout et al., 2018).

This paper is structured as follows. Section 2 describes and compares the three different QC methods. Section 3 provides instructions and guidelines on how to run these QC methods in the OpenSense sandbox environment. In Section 4, a case study where these QC methods have been applied using a PWS dataset from the Amsterdam region in the Netherlands is presented. This is followed by a discussion and conclusions and recommendations for the usage of these QC methods in sections 5 and 6, respectively.

## 2 Description of the QC algorithms

### 2.1 PWSQC

PWSQC was originally developed and published by de Vos et al. (2019). It consists of several QC modules, all relying on neighbour checks. Neighbours are defined as all PWS within a spatial range, which is a parameterized value. The range should reflect the distance over which one assumes neighbouring PWS to capture similar rainfall dynamics. This value needs to be chosen carefully for the local climate and the temporal resolution of the PWS data, as the rainfall fields corresponding with longer timesteps are more homogeneous than those from short timesteps (Terink et al., 2018). For high temporal resolution PWS data, neighbour comparisons can only be sensible if neighbour PWS are selected with a short range. As the selection is only based on distance, local effects like elevation or surface use are ignored. Another parameter in the method sets the required minimum number of neighbour PWS that provide observations each timestep, to ensure the neighbour comparison is robust. When this parameter is chosen too high, in sparse areas of the network the minimum number of neighbour observations is never reached.

A Faulty Zero (FZ) filter checks for periods of 0 mm rainfall where the median of nearby PWS observations is above zero. The High Influx (HI) filter detects unrealistic high measurements compared to its surroundings by comparing values against the median of its neighbour PWS, with a fixed threshold for low rainfall intensities and a dynamic high threshold during rainfall events (as measured by nearby PWS). The Station Outlier (SO) check calculates the correlation between a PWS and each neighbouring PWS and starts flagging when the correlation of most becomes too low. Finally, a dynamic bias correction factor (which differs for each PWS and can change in time) is calculated and applied to the observations. For the initial value of the bias correction factor, an auxiliary dataset can be considered to derive a proxy for the overall bias of the whole dataset. This will improve the results, but no auxiliary data is required for the application of PWSQC. The method attributes flags to individual observations that can then be filtered, it does not exclude complete time-series. PWSQC has originally been applied on the same dataset as the one used in this paper and showed promising results. The method has been implemented in R, and is openly available (de Vos, 2021). Later, a radar version of this algorithm has been constructed in Python that makes use of unadjusted radar data at the location of a PWS as input for the QC. Then neighbouring PWS are only employed to improve the radar input data (Van Andel, 2021).

### 2.2 PWS-pyQC

PWS-pyQC was first introduced by Bárdossy et al. (2021). It was used in a German-wide study in Graf et al. (2021) and an event-based analysis in Bárdossy et al. (2022) showing the potential of PWS data to improve precipitation interpolation. The method is implemented in Python and is open-source software available in El Hachem (2022). The QC algorithm consists of three main modules : the first identifies reliable PWS using a space-time dependence structure derived from a reference observation network (denoted as primary network). The main assumption is that the PWS values might be wrong but their relative order (i.e. their ranks) are correct. First, the indicator correlation values are calculated from the reference network and the PWS observation series individually. Each time series is transformed into a binary series depending on the corresponding precipita-

tion value ($z = F^{-1}(\alpha)$) for the selected quantile value ($\alpha$) which has to be chosen depending on the temporal resolution of the data (c.f. Bárdossy et al., 2021). For hourly values, the indicator series are obtained by using a threshold of $\alpha = 0.99$. After deriving the pair-wise indicator correlation matrix from the reference network, the PWS data can be filtered. For each PWS, the indicator correlation with the nearest neighboring primary station is calculated and compared, for the same separating distance, to the corresponding value in the reference correlation matrix. This allows for identifying and filtering PWS observation series that do not fit in the reference correlation structure. An advantage of using indicator correlations is that the absolute values do not matter (for example, if 50 mm and 10 mm both exceed the threshold, $z = F^{-1}(\alpha)$ they are both transformed to the value of 1). Furthermore, stations with incorrect location information can be identified as well. A disadvantage of this approach is that the complete time series of the corresponding PWS is disregarded and filtered out.

The second module corrects the bias in the magnitudes of the values of each PWS individually using the ranks of the PWS and the corresponding neighboring primary observations. To that end, for every PWS value larger than 0 mm, its corresponding rank and subsequent quantile are identified. For the same quantile level, the corresponding precipitation values at the nearest primary stations are identified. These are then used to interpolate the precipitation value at the PWS location. This corrects the bias in the PWS values individually while preserving their ranks. It is the most time-consuming part of PWS-pyQC as each hourly value has to be individually corrected.

The third module is an event-based filter to identify erroneous PWS observations (false zeros, false extremes) for corresponding time intervals. The filter is based on a leave-one-out cross-validation approach. After applying a Box-Cox transformation to transform the PWS and primary data, each PWS value (after bias correction) is removed from the dataset and is re-estimated using the observation from the primary network. The ratio between the absolute difference of the estimated and observed values and the kriging estimation variance is noted. Large ratios indicate that the observation is an outlier (a single value or a false measurement). Depending on the magnitude of the flagged observation (zero or high precipitation value) the user has to decide to keep or disregard the value. For this step, external information such as weather radar data or discharge value (for headwater catchments) could be used to distinguish between a false measurement and a single event. Note that this filter was further developed in El Hachem et al. (2022).

PWS-pyQC relies on a reference network (a primary network) with reliable observations to filter the PWS data. This is usually acquired from the official rain gauge network. However, in the study area, there is only one KNMI rain gauge with hourly temporal resolution available, which is not sufficient for deriving a reference dependence structure. Hence, this dependence structure was derived from a radar gauge-adjusted KNMI product by taking the times series of 20 randomly chosen pixels as the primary network. A sensitivity analysis will help determine what the impact might be when 20 other pixels (or even a different number of pixels) are chosen as primary network proxy. If an official rain gauge network is available with enough gauges within the study area it is recommended to use it.

## 2.3 GSDR-QC

GSDR-QC is the QC algorithm developed to construct the Global SubDaily Rainfall dataset (Lewis et al., 2019), and is fully described in Lewis et al. (2021). The algorithm flags and removes suspicious individual observations, rather than entire gauge

datasets, and does not attempt to alter (bias correct) observations, making it the most conservative of the QC methods described herein. The GSDR-QC applies user-defined thresholds for hourly and daily maximum rainfall (appropriate to the extent of the study area), nearest neighbour checks and uses climate indices defined by the Expert Team on Climate Change Detection Indices (ETCCDI, https://www.ecad.eu/indicesextremes/) comprising R99p, PCPTOT, Rx1day, CDD and SDII. These are described in Table 1 of Lewis et al. (2021). The outputs of the GSDR-QC allow the user to evaluate QC summary overviews to establish where faults lie, offering insight into the type of errors.

The complete procedure relies on a two-step process; first flagging suspicious observations, followed by the application of a rule base that uses the flags to remove unreliable observations. The process is comprehensive, addressing all WMO tests recommended for rainfall QC including; format; completeness; consistency; tolerance/range; and, spike and streak (WMO, 2021). A reference table describing each test is presented in the Supplementary Information of (Villalobos-Herrera et al., 2022). There are 25 QC checks that flag suspicious data comprising QC1 - QC7 that identify where a substantial portion of the gauge data appears to be suspicious (i.e. the gauge is seemingly unreliable), QC8 - QC11 that flag suspiciously high values, QC12 that flags long periods without rainfall and QC13 - QC15 that flag suspect accumulations or repeated values. Checks QC8, QC9 and QC11 - QC15 use ETCCDI indices as reference data (QC10 implements the user defined maximum rainfall values). There are then further QC flags applied based on observations from neighbouring gauges including QC16 - QC25 that flag mismatches between neighbours including high rainfall, dry periods and the timing of rainfall. In the original implementation, three of these checks require access to the restricted-access Global Precipitation Climatology Centre (GPCC) daily and monthly precipitation databases; however, these are not essential and are not used in this 'local' implementation of the GSDR-QC algorithm (as applied herein).

Once data have been flagged, a rule base uses eightof the 25 QC checks to determine suspicious observations which are removed from the dataset. Briefly, the rule base applies uses the QC checks against neighbouring gauges ($\times$2), checks for extremely large values ($\times$2), for long dry spells, for repeated non-zero values, and for suspect daily and monthly accumulations. Table 3 in Lewis et al. (2021) provides a full description of the rule base. A key aspect of the neighbour checks is that they are applied to an aggregated daily total, to avoid any potential issue caused by the higher variability and intermittency of hourly rainfall (Lewis et al., 2021). This variability and intermittency tend to be higher in data originating from official networks (the original application of GSDR-QC) since they are much less dense than PWS networks, especially in urban areas (O'Hara et al., 2023).

The GSDR-QC can be tailored to the dataset/location in many ways, the most obvious being defining an appropriate maximum hourly and daily threshold for rainfall and required for the extreme value checks, and the determination of "nearest neighbours". In our case study we opted for hourly maximum rainfall (90.7 mm) and daily maximum (131.7 mm) that were representative for the climate of the study area. We allowed inter comparison between up to 10 nearest neighbours that were within a 50 Km radius and had a minimum of 1 year of overlapping data.

**Table 1.** Technical overview of the QC algorithms

| | PWSQC | PWS-pyQC | GSDR-QC Local |
|---|---|---|---|
| QC modules | 1. Neighbour selection<br>2. Faulty Zeroes & High Influx filter<br>3. Station Outlier filter & bias correction factor determination | 1. Indicator based filter<br>2. Bias correction<br>3. Event based filter | 1. Flagging of suspicious observations using defined rule base<br>2. Filtering of suspicious observations not meeting QC criteria |
| Reference dataset required | No, but optional part of initialization of bias correction factor determination | Yes, required for 1, 2 and 3 | Yes, ETCCDI data plus user defined maximum daily and hourly thresholds |
| Programming language | R | Python | Python |
| Ground truth used in method | Median values from neighbouring PWS | PWS should fit in space-time dependence structure of reference data | Neighbouring gauges are compared to each other and optionally against a reference dataset |
| Level of QC-allocation | - Per measurement | - Per full PWS time series<br>- Event based | - Per individual measurement<br>- Dynamic nature is suitable for longer time series |
| Output after running QC method | - Original PWS dataset<br>- 3 flag files conveying flag attribution to individual observations for all three QC<br>- 1 file with bias correction factors generated for each observation<br>- Bias adjusted PWS dataset with only reliable observations | - Set of trustworthy PWS<br>- Individual bias correction for each time series<br>- Implausible time intervals removed for each time series | - Flag file for each gauge showing individual test results<br>- Output file with reliable observations |
| QC methods are available in OpenSense sandbox (https://github.com/OpenSenseAction/OPENSENSE_sandbox) | | | |

## 2.4 Overview of QC technical and operational guidelines

Table 1 recaps the technical differences between the three QC methods. Table 2 offers those interested to apply QC on PWS data an overview on the applicability of the three different QC methods, thus supporting them in choosing the most suitable QC method for processing specific datasets (time series length, temporal resolution, PWS network density etc.), the availability of a reference dataset and computational resources.

## 3 Getting started in the sandbox

All three QC algorithms are available under https://github.com/OpenSenseAction/OPENSENSE_sandbox which is the OpenSense GitHub repository. This repository includes a binder which allows users to run and explore the code online as well as instructions on how to install the environment locally. It is not required to have a GitHub account.

The version of the PWSQC code available in the OpenSense sandbox (https://github.com/OpenSenseAction/OPENSENSE_ sandbox/tree/main/PWSQC_R_notebook) is practically identical to the originally published R-code, with an added introduction

to download a PWS dataset directly from the internet repository and save it in the correct format to get started. The code runs in a R-kernel, and all steps are listed in two Jupyter Notebooks. Due to the time it takes the code to run in the sandbox, particularly

**Table 2.** Operational guidelines for the use of the QC algorithms

| Applicability regarding | PWSQC | PWS-pyQC | GSDR-QC Local |
|---|---|---|---|
| Temporal scale | HI-filter has no lead-up time, but (with default parameters) FZ filter requires 30 min and SO-filter and bias correction require $\geq$2 weeks of data with >100 nonzero intervals. Most suitable for long periods of continuous data. | Time series should be long enough to include significant number of rain events, which is dependent on the climatic region and temporal resolution. | Where neighbouring PWS are available within 50 km there is a minimum requirement of 1 year of overlapping data. Otherwise, where climate indices are available 1 month minimum of data is required. |
| Spatial scale | Network can span large areas, provided that neighbour PWS values are a good proxy of the ground truth throughout the network. Neighbours are defined by a range around a station which assumes climatological agreement with neighbours in all directions | Due to need for reference set, PWS network has to overlap with reference network. For the indicator filter, the data from the reference network needs to represent the local spatial and temporal rainfall variability, but a temporal overlap is not necessary. | There are no limitations to the spatial scale. Consideration must be given that the same daily and hourly maximum rainfall thresholds are applied on the whole area. |
| Temporal resolution | 5 minute timesteps (or longer) | 1 hour timesteps (or longer) | 1 hour timesteps (or longer) |
| Spatial resolution | Due to neighbour checks, most suitable for dense networks | Applicable for both dense and sparse networks | Applicable for both dense and sparse networks |
| Operational potential | Current version of code works only on static dataset, but theory applies for operational application | Current version of code works only on static dataset, but theory applies for operational application | Developed for static datasets |
| Approximated runtime for Amsterdam PWS dataset | 1. Neighbour selection: flash 2. FZ and HI filter: lunch break 3. SO and bias correction: weekend | 1. Indicator correlation: flash 2. Bias correction: lunch break 3. Event Filter: coffee time | 1. Create gauge objects: flash 2. Run QC: coffee time 3. Extract QC summary: flash |
| Impact of PWS network scaling on run time | As whole network needs to be evaluated for each timestep, large dependency on number of stations | Calculation of distance matrices increases nonlinear with number of stations | Linear with number of PWS |

the step with the SO filter and the dynamic bias correction factor calculation, users may explore the minimal example first, where 10 timeseries in the PWS dataset are attributed FZ and HI flags only, and the results are visualized.

The PWSpy-QC repository includes a folder with a Jupyter notebook showcasing the workflow of the algorithm with the Amsterdam PWS dataset. After the import of the modules which includes the code for the PWS-pyQC modules in the file PWSpyqcFunctions.py, some user specific settings like the maximum distance for which the indicator correlation is calculated or the threshold percentile for the indicator correlation can be made. The user can set or change the three parameter values of the QC, which need to be adjusted according to the temporal resolution of the data and the network density. The sample data can be loaded and the notebook then produces several plots showing the locations of the primary stations and PWS. This is followed by examples for the different filters. The indicator filter accepted and rejected PWS that do not match the spatial correlation pattern of the reference network. The bias correction of individual PWS are also showcased in the notebook. The bias correction described in Bárdossy et al. (2021) is based on a quantile mapping between the PWS and the surrounding primary station. This bias-correction takes about 2 h for the Amsterdam PWS dataset. Optionally, the bias corrected PWS data can be saved as .csv file. The event filter is applied after the bias correction and requires few minutes execution time. For every timestep, the filter flags individual PWS whose values deviate too much from the surrounding reference network values. The output of this filter can be saved as .csv file as well.

The GSDR-QC repository includes the scripts required to prepare data for implementing the QC, running the QC algorithm and for generating summary outputs on the impact of the QC on the observations from each PWS. Where user defined mod-

ifications can/need to be made the scripts are available as Jupyter notebooks. Scripts with functions used in GSDR-QC are provided as .py scripts. There is a step-by-step guide to support users, which highlights how to apply the changes for localisation (locally appropriate maximum hourly and daily rainfall, and duration of overlap of neighbouring observations). The data preparation script is provided as an example, as the exact process will be determined by the original format of the PWS observations.

We provide an example of how to implement the three QC algorithms in the case study, and highlight some considerations and limitations users should be aware of when selecting the most appropriate method.

## 4 Case study

The three QC methods have been applied on the same PWS dataset of 25 months spanning the Amsterdam metropolitan area in the Netherlands. For details on this dataset we refer to Figure 1, de Vos (2019); de Vos et al. (2019) and the OpenSense sandbox. Figure 1 highlights that the spatial spread of PWS is typically not homogeneous and there are also edge effects by evaluating only PWS within a bounded box. As the PWS-pyQC algorithm and the iteration of GSDR-QC available via the OpenSense sandbox require at least a 1-hour temporal resolution as data input, the 5-min PWS dataset has been aggregated to hourly values. Hourly values were only constructed with the completeness condition that at least 10 out of 12 intervals were available, otherwise the value became 'NA'. PWSQC has been applied to the PWS dataset in its raw 5-min temporal resolution. Intervals that were allocated a FZ, HI, and/or a SO error were excluded. After QC, the PWS dataset was aggregated to hourly values with the same completeness condition.

As a reference dataset, a gauge-adjusted radar product from the Royal Netherlands Meteorological Institute (KNMI) with a 1 km spatial and 5-minute temporal resolution was used (referred to as radar reference from here on). It is a climatological product of radar rainfall depths corrected with validated hourly automatic rain gauge measurements and validated daily manual rain gauge measurements, constructed with a considerable delay (i.e. not real-time available), available on the KNMI data platform (KNMI, 2023). Additional details on its construction can be found in Overeem et al. (2009a, b, 2011). It may seem contradictory to consider a radar product as ground truth, while making the case that PWS data may significantly improve these existing rainfall measurement techniques. Note that this radar product combines three types of rainfall information (two radars and two gauge networks, one automatic and one manual) with a significant delay, resulting in it being the best available reference to work with. The benefit of PWS is its high spatial density and availability in real-time. By merging radar with PWS, the resulting product combines the best from both techniques, see for example Overeem et al. (2024); Nielsen et al. (2024); Overeem et al. (2023).

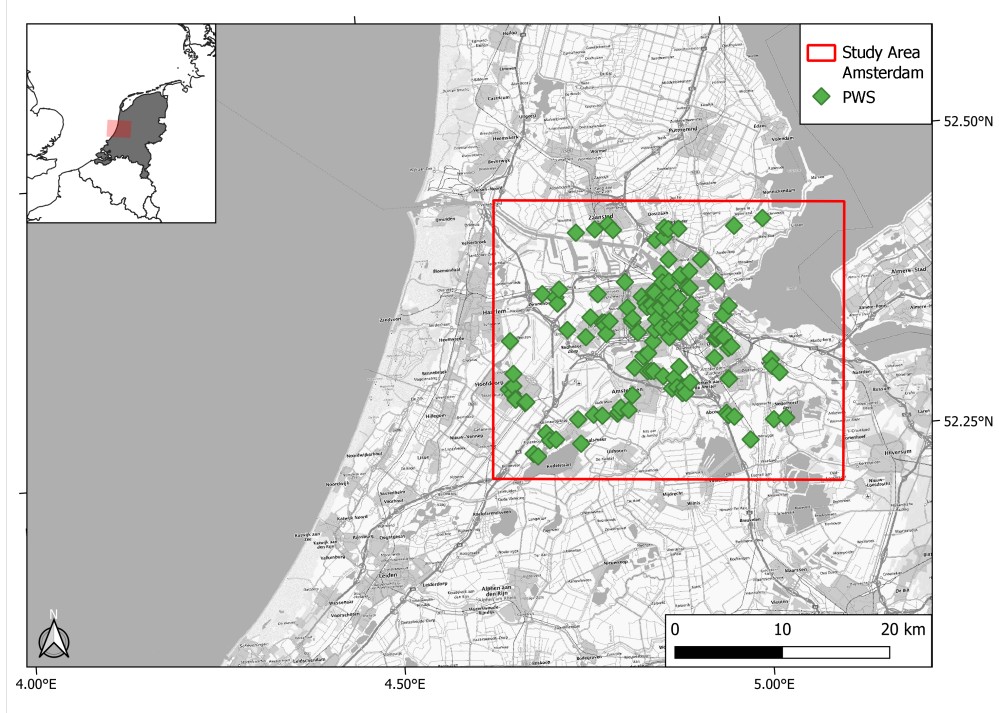

**Figure 1.** Map of the study area in the Amsterdam metropolitan area in the Netherlands. The red box in the map and overview map depicts the domain used for the QC comparison. The PWS locations are denoted by the green dots. Background map: WMS TopPlusOpen (https://gdk.gdi-de.org/geonetwork/srv/api/records/8BDFB79F-A3FD-4668-88D3-DFD957F265C2).

The results highlight four distinct 24-hour rainfall events, selected to represent different spatiotemporal rainfall characteristics. The events were chosen because the majority of PWS recorded significant rainfall over an extended duration, making them appropriate for the application of quality control (QC) algorithms.

Figure 2 shows the Ordinary Kriging interpolated rainfall maps on a ≈ 1km grid after the QCs have been applied to the PWS data and the radar reference for Event 4 (2018-05-29 08:00 – 2018-05-30 08:00 UTC). The figures corresponding to the other 3 events, details on the interpolation method and the difference maps of the four events can be found in Appendix B. The highest peaks in the radar reference are not captured by PWSQC. The rainfall in the southwest part of the area, where the airport is located and PWS density is low (see Fig. 1), is underestimated by all, but most severely by GSDR-QC which is the

least sensitive to remove faulty zeroes in the data. PWS-pyQC has the best metrics for this event although only 50% of the PWS are retained on average (c.f. Table D1). The rainfall maps after applying each of the QC algorithms show similar patterns to the radar reference.

GSDR-QC shows the most remaining data after QC, while PWS-pyQC rejects most PWS stations on average. This is related to the faulty zero checks in the other two methods that are implemented at the sub-daily timescale, whereas the GSDR-QC

applies the check to daily aggregated data, resulting in reduced sensitivity to missing observations (see also scatter plots in

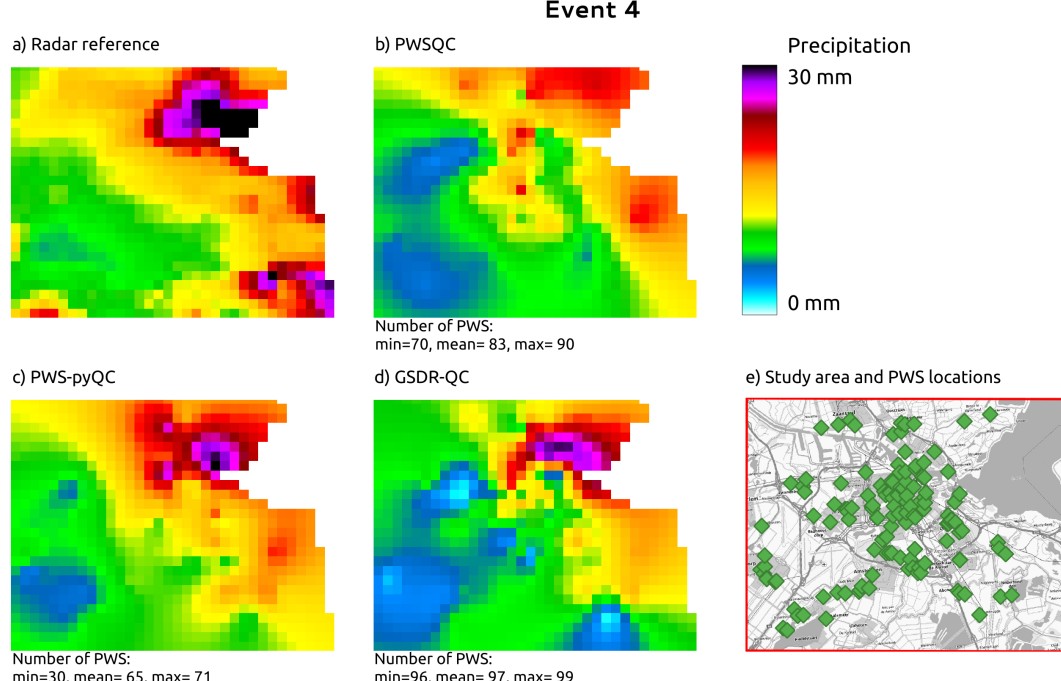

**Figure 2.** Rainfall maps for event 4. Panel a) shows the gauge-adjusted radar accumulation. Panels b), c), and d) show the interpolated PWS accumulations using the QC algorithms *PWSQC*, *PWS-pyQC* and *GSDR-QC*, respectively. Panel e) shows the locations of all PWS. Under each map the data availability after QC is indicated by providing the number of PWS with hourly data, that were used to generate interpolated maps for the hour with the fewest (min) and highest (max) PWS remaining after QC, as well as the average (mean) over the 24-hour maps. Background map in panel e): WMS TopPlusOpen (https://gdk.gdi-de.org/geonetwork/srv/api/records/8BDFB79F-A3FD-4668-88D3-DFD957F265C2).

Appendix B). Hence, and because no bias correction is implemented in GSDR-QC, the tendency of a higher negative bias was to be expected. Results after PWS-pyQC yield similar values for bias and Pearson correlation as PWSQC, and values for the coefficient of variation smaller than the other two QC methods.

Figure C1 shows an example of the hourly rainfall areal averaged over the full domain for event 4. The spatially averaged rainfall after all methods have been applied approximates the radar reference well, with the least underestimation in PWSQC and PWS-pyQC. Even though the spatial rainfall maps present large differences as seen in Figure 2, this is largely averaged out over the domain. More results regarding the four events can be found in Appendix B, C and D. These results provide insight in the number of remaining data, correspondence with radar reference regarding spatial patterns, areal averages and overall performance metrics for the study area. The relative performance should be interpreted as indicative, as they do not constitute a complete benchmark study.

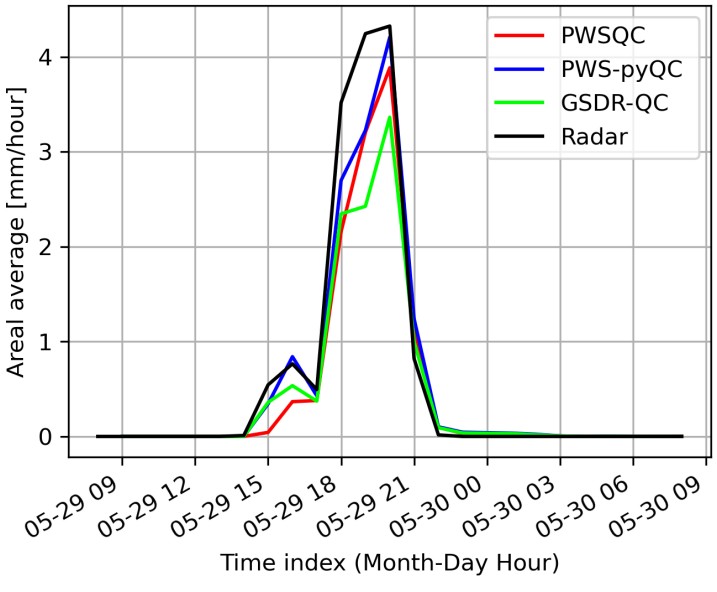

(a) Areal average for event 4

**Figure 3.** Example of areal rainfall over the Amsterdam metropolitan area for event 4.

## 5 Discussion

Within this technical note, a guideline to using three different QC algorithms was presented. Interested users have the chance to familiarize themselves with each individual QC and gain insight into their use. The information presented in Table 1 provides a comprehensive overview of the main differences between the 3 QC algorithms. This information should be beneficial for
new users who are interested in using the QC algorithms. For instance, PWSQC does not require any additional information from more reliable observations and can thus be used in areas without reference data where only PWS data is available. PWS-pyQC requires a reference dataset (primary stations) set to derive information about the spatial pattern of indicator correlations and to apply the other filters. Such a dataset can either be a dense rain gauge network or as shown in this study, a gauge-adjusted radar product. In the absence of such a dataset, PWS-pyQC cannot be used. GSDR-QC requires a reference dataset
of gridded precipitation datasets and user defined maximum rainfall thresholds. PWS-pyQC typically retains the smallest number of stations compared to PWSQC and GSDR-QC. As the indicator correlation filter of PWS-pyQC rejects the complete PWS series whereas the other QC methods flag and/or remove suspicious individual observations. PWSQC has been applied conservatively where if not enough data was available to determine a flag, the data is not excluded. Given that PWSQC is applicable to 5 min time series and PWS-pyQC and GSDR-QC to hourly, the calculation of number of remaining observations
is slightly different.

In Table 2 several operational guidelines are provided. Such information is beneficial to select the appropriate QC for the given data availability. The availability of the three QC algorithms within the OpenSense sandbox along with the data from

the case study enables testing and experimenting with each QC. Moreover, the users can easily modify the QC parameter values without the need to change the main QC functions. Within the case study, the three QC algorithms were applied to
255 the same dataset. An interpolation for one daily event showed that all QC algorithms are able to adequately estimate the average areal rainfall, although the spatial patterns can largely differ. This preliminary analysis cannot provide a detailed comparison between the QC performances. For this, a sensitivity analysis regarding the choice of parameters and reference data would be needed. In addition, long record periods and different climatic conditions would be needed. Such an analysis is beyond the scope of this technical note, as the main aim is to provide interested users in guidelines for using the different QC
algorithms. Each QC algorithm was developed and presented in original works, where the validity of the QC algorithm was tested (de Vos et al., 2019; Bárdossy et al., 2021; Lewis et al., 2019). The PWS dataset used in this study is of a relative small size. Upscaling the QC algorithms for larger datasets, e.g. covering Europe (Netatmo (2021): EUMETNET Sandbox, 2021) requires additional steps. For instance, the PWS-pyQC applies the filters for every PWS independently, hence a parallelisation of the filters allows handling large datasets and time consuming steps. PWSQC cannot be parallelized per time subset due
to lead-up time. Parallelization per subset of stations is possible, but the whole PWS dataset needs to be within the working memory of each parallel run to ensure that a PWS's neighbours are always part of the analysis. An alternative approach is used in Overeem et al. (2024), and in the application of spin off code from (Van Andel, 2021) on Dutch water board gauge data, to apply the FZ and HI filters only, as these are more efficiently run than the SO filter and bias correction factor allocation. GSDR-QC is easily run in parallel as each rain gauge is analyzed in a separate process and multiple gauges may be analyzed
simultaneously. The Python code has be written to be efficient and the whole case study sample is processed in a few minutes on an 8-core laptop. The GSDR-QC is therefore the fastest to implement as it was designed for the quality control of a global dataset.

## 6 Conclusions and outlook

In this work, we presented a guideline to using open-source QC algorithms for PWS rainfall data based on a single dataset and
275 a case study. The aim was to provide an example of how QC for PWS can be used and to contextualise the additional input data requirements and the technical and operational guidelines for the individual QC methods. Interested users can select the most appropriate QC algorithm for their case study, and whilst the subsequent dataset might not be perfect, there is an improvement from the raw data. Studies like the ones from Bárdossy et al. (2022) and Overeem et al. (2024) have shown the added value of PWS for improving rainfall estimates for extreme events in Germany and quantitative precipitation estimation on a European
scale, respectively.

In our example all presented QC methods improve the quality of PWS rainfall data, however, this single example does not provide sufficient data to accurately benchmark the three algorithms. Additional work is required for comprehensive sensitivity testing across a range of environments, monitoring networks, and weather patterns to provide more quantitative guidance on the most appropriate QC method.

We plea for making open opportunistic sensing data on a European or even global level (or restricted access for research purposes), which would foster the development and improvement of QC and rainfall retrieval algorithms. Eventually, this will lead to improved precipitation products and applications such as validation of weather/climate models, hydrological modelling, nowcasting, etc. Furthermore, there is a need for large benchmark radar and rain gauge datasets from different regions and climates. Such benchmark datasets would facilitate a fair intercomparison of QC algorithms and even different opportunistic sensor rainfall estimates from commercial microwave links (CMLs) and satellite microwave links (SMLs). Intercomparison studies also require appropriate metrics and the aforementioned datasets. A discussion on standardized benchmark metrics to be used for intercomparison studies is needed. Benchmarking and intercomparison of algorithms for opportunistic sensor data, merging of opportunistic sensor data and traditional data from rain gauges and radars, and the integration of these data into standard observation systems are objectives that are currently being addressed in the OpenSense COST Action (https://opensenseaction.eu/).

An ongoing activity within working group 2 of the OpenSense COST Action is the open-source software implementation the QC algorithms for processing OS data. The aim is develop a Python module which includes all of the modules of the QC algorithms presented in this paper. In the long run this will replace the current QC algorithms in the OpenSense sandbox and allow users to apply and even combine these in a uniform standardised programming language.

*Data availability.* The gauge-adjusted radar product from the Royal Netherlands Meteorological Institute (KNMI) is freely available on the KNMI data platform: https://dataplatform.knmi.nl/dataset/rad-nl25-rac-mfbs-5min-netcdf4-2-0 (KNMI, 2023). The employed PWS dataset is publicly available: https://doi.org/10.1029/2019GL083731 (de Vos, 2019)

*Code and data availability.* The corresponding code for the analysis is available upon request from the contact author. All QC software is available as open source and can be accessed in the OpenSense sandbox (https://github.com/OpenSenseAction/OPENSENSE_sandbox) in addition to their original locations **PWSQC** is available as R code under https://github.com/LottedeVos/PWSQC. **PWS-pyQC** is available as Python code under https://github.com/AbbasElHachem/pws-pyqc. **GSDR-QC** is available as Python code under https://github.com/nclwater/intense-qc and https://pypi.org/project/intense/.

*Author contributions.* AEH implemented PWS-pyQC to the raw PWS data and did the analysis following the outputs of the 3 QC methods. JS assisted in analysing and displaying the results. TOH and RHV implemented GDDR-QC to the raw PWS data. AO, RU, and AB aided in reviewing and structuring the manuscript. LdV organized the workflow and implemented PWSQC to the raw PWS data. All authors contributed to writing the manuscript.

*Competing interests.* At least one of the (co-)authors is a member of the editorial board Hydrology and Earth System Sciences.

*Acknowledgements.* This work was supported by the COST Action Opportunistic Precipitation Sensing Network (OpenSense; CA20136; https://opensenseaction.eu/). We thank the private company Netatmo for providing the PWS dataset and all citizen scientists for buying, installing, and maintaining these stations. We thank Georges Schutz and one anonymous reviewer for their valuable comments that improved the quality of our manuscript.

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

**Table A1.** Overview of the four rainfall events chosen for this case study

|  | Start | End | Event Characteristics |
|---|---|---|---|
| Event 1 | 12 May 2017 8:00 UTC | 13 May 2017 8:00 UTC | Several showers |
| Event 2 | 27 Nov 2017 8:00 UTC | 28 Nov 2017 8:00 UTC | Homogeneous rainfall, dry spell from 22:00 to 05:00 |
| Event 3 | 15 Jan 2018 8:00 UTC | 16 Jan 2018 8:00 UTC | Homogeneous rainfall, one very evident outlier PWS |
| Event 4 | 29 May 2018 8:00 UTC | 30 May 2018 8:00 UTC | Convective rainfall from 14:00 to 22:00 |

## Appendix A: Metrics for all four events

For the four events, 24h accumulations maps based on hourly interpolations were derived and compared. Subsequently, hourly areal averages over the study domain were calculated and compared. Furthermore, the number of remaining stations was calculated. A point value comparison was done by calculating the pair-wise correlation value, the bias and the coefficient of variation (CV) between the PWS and reference data for all PWS locations in the study domain.

The first evaluation metric is the Pearson correlation. It is a widely used pair-wise dependence measure to identify the presence (or absence), the strength, and the direction of a linear relationship between pairs of variables (for example, $x$ and $y$). The equation for calculating the Pearson correlation can be seen in equation A1.

$$r_{xy} = \frac{\sum_{i=1}^{n}(x_i - \bar{x})(y_i - \bar{y})}{\sqrt{\sum_{i=1}^{n}(x_i - \bar{x})^2}\sqrt{\sum_{i=1}^{n}(y_i - \bar{y})^2}} = \frac{cov(X,Y)}{\sigma_X \sigma_Y} \tag{A1}$$

The second metric is the relative bias defined as follows (eq. A2).

$$Bias = \frac{\overline{(x-y)}}{\bar{y}} \tag{A2}$$

The third metric is the coefficient of variation (CV, eq. A3) and is used to quantify the dispersion in the data.

$$CV = \frac{\sigma(x-y)}{\bar{y}} \tag{A3}$$

Where:

$x =$  variable to evaluate

$y =$  reference variable

$\sigma =$  standard deviation

$r_{xy} =$  Pearson correlation coefficient

$x_i =$  value of $x$ at time interval $i$

$\bar{x} =$  average value of time series $x$

$y_i =$  value of $y$, the reference, at time interval $i$

$\bar{y} =$  average value of time series $y$, the reference

$n =$  number of observations

## Appendix B: Additional Rainfall Maps

Figures A1-A3 show the interpolated 24h rainfall map after the corresponding QC algorithms have been applied. Figures A4-A7 the filtered and corrected PWS data are interpolated using Ordinary Kriging (OK) on the same grid as the reference dataset. OK utilizes the spatial configuration of the points which is quantified by a fitted variogram model. The latter is derived in the rank space domain following the procedure in Lebrenz and Bárdossy (2017). The parameters (sill and range) were further adapted to adhere with the bounds and order of magnitudes with those derived for the Dutch conditions in the work of Van de Beek et al. (2012). In case no suitable variogram could be derived, for example, due to the large number of zeros, an average spherical variogram was used, without a nugget value and with a sill scaled according to the data variance. For every hour of the selected daily event with positive PWS observations, the values in the domain are spatially interpolated. The number of accepted PWS is accordingly noted. The daily map is acquired by accumulating the hourly maps.

For event 2 (Fig. A2), all QC methods show more spatial variability than the radar reference, which is caused by some faulty zeros which are not detected by PWSQC and GSDR. Also, some higher values appear which were obviously not identified as outliers by the QC methods. For event 3 (Fig. A2), there is a very evident outlier PWS with high rainfall amounts over 30 mm. This outlier was not detected by PWSQC and GSDR.

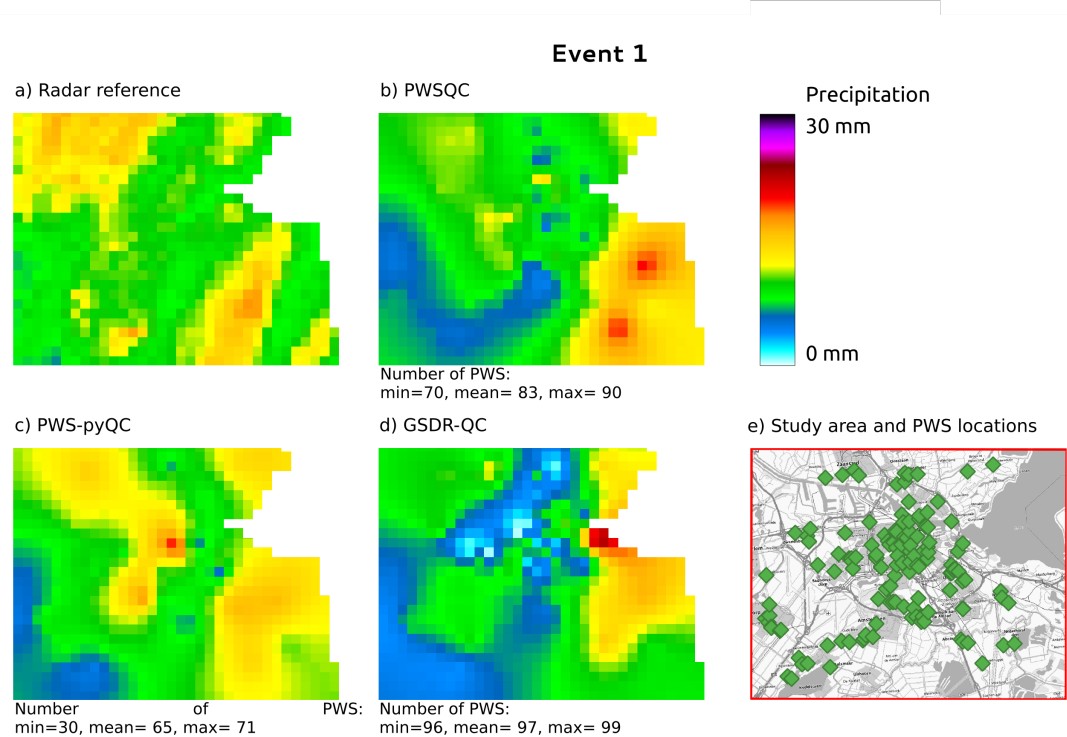

**Figure A1.** Rainfall maps for event 1. Panel a) shows the gauge-adjusted radar accumulation. Panels b), c), and d) show the interpolated PWS accumulations using the QC algorithms *PWSQC*, *PWS-pyQC* and *GSDR-QC*, respectively. Panel e) shows the locations of all PWS. Under each map the data availability after QC is indicated by providing the number of PWS with hourly data, that were used to generate interpolated maps for the hour with the fewest (min) and highest (max) PWS remaining after QC, as well as the average (mean) over the 24-hour maps. Background map in panel e): WMS TopPlusOpen (https://gdk.gdi-de.org/geonetwork/srv/api/records/8BDFB79F-A3FD-4668-88D3-DFD957F265C2).

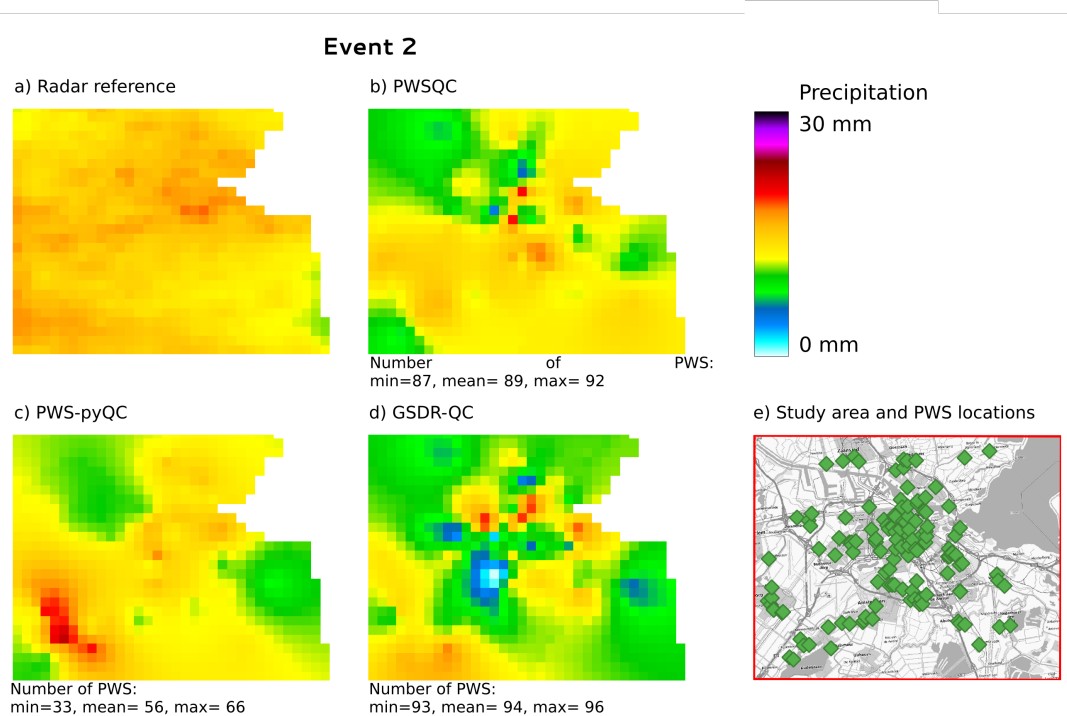

**Figure A2.** Rainfall maps for event 2. Panel a) shows the gauge-adjusted radar accumulation. Panels b), c), and d) show the interpolated PWS accumulations using the QC algorithms PWSQC, PWS-pyQC and GSDR-QC, respectively. Panel e) shows the locations of all PWS. Under each map the data availability after QC is indicated by providing the number of PWS with hourly data, that were used to generate interpolated maps for the hour with the fewest (min) and highest (max) PWS remaining after QC, as well as the average (mean) over the 24-hour maps. Background map in panel e): WMS TopPlusOpen (https://gdk.gdi-de.org/geonetwork/srv/api/records/8BDFB79F-A3FD-4668-88D3-DFD957F265C2).

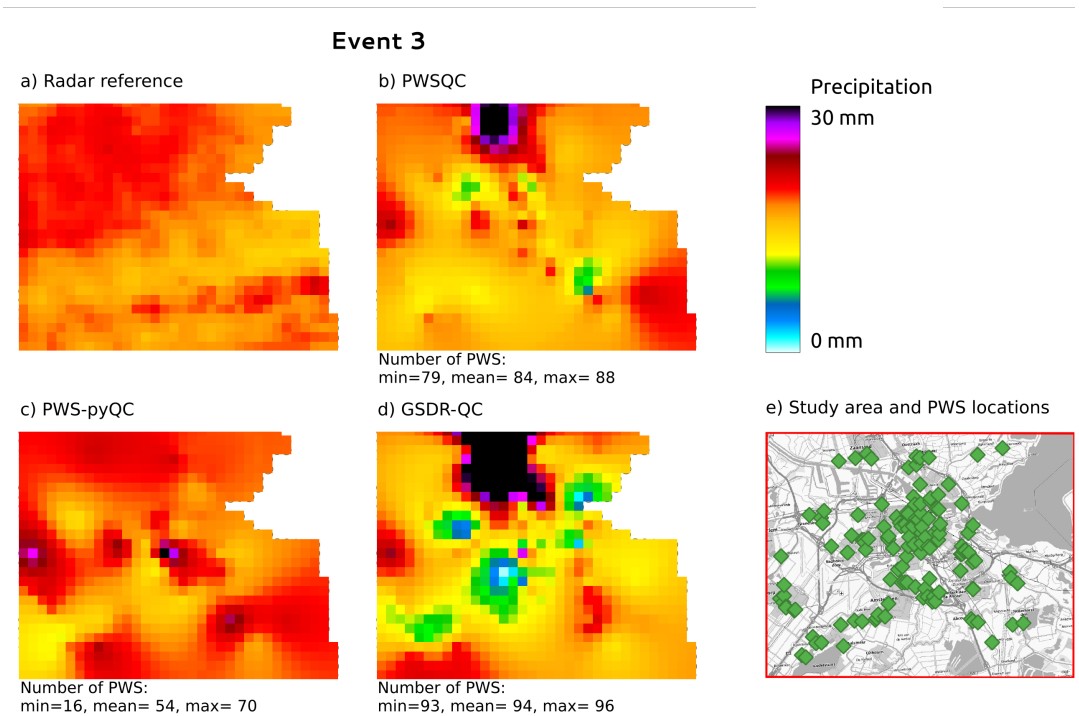

**Figure A3.** Rainfall maps for event 3. Panel a) shows the gauge-adjusted radar accumulation. Panels b), c), and d) show the interpolated PWS accumulations using the QC algorithms *PWSQC*, *PWS-pyQC* and *GSDR-QC*, respectively. Panel e) shows the locations of all PWS. Under each map the data availability after QC is indicated by providing the number of PWS with hourly data, that were used to generate interpolated maps for the hour with the fewest (min) and highest (max) PWS remaining after QC, as well as the average (mean) over the 24-hour maps. Background map in panel e): WMS TopPlusOpen (https://gdk.gdi-de.org/geonetwork/srv/api/records/8BDFB79F-A3FD-4668-88D3-DFD957F265C2).

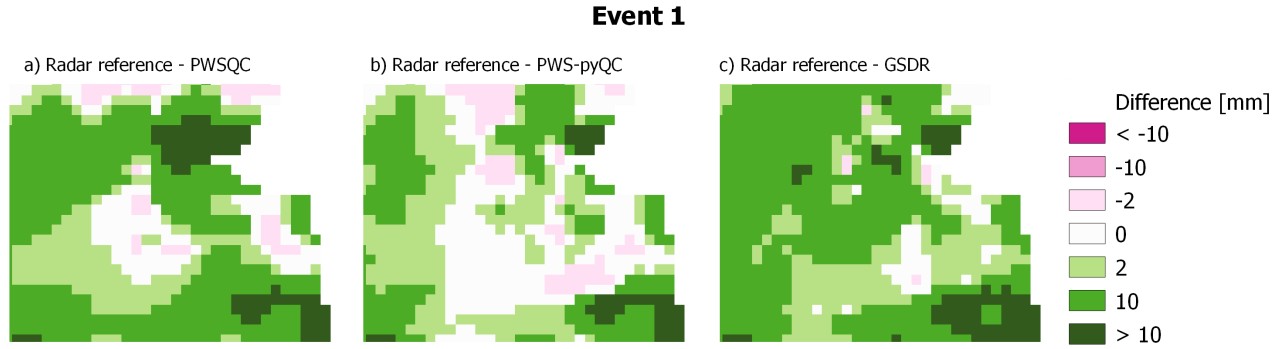

**Figure A4.** Differences between the radar reference and the interpolated maps from the three QC algorithms for event 1.

**Event 2**

a) Radar reference - PWSQC  b) Radar reference - PWS-pyQC  c) Radar reference - GSDR

Difference [mm]
< -10
-10
-2
0
2
10
> 10

**Figure A5.** Differences between the radar reference and the interpolated maps from the three QC algorithms for event 2.

**Event 3**

a) Radar reference - PWSQC  b) Radar reference - PWS-pyQC  c) Radar reference - GSDR

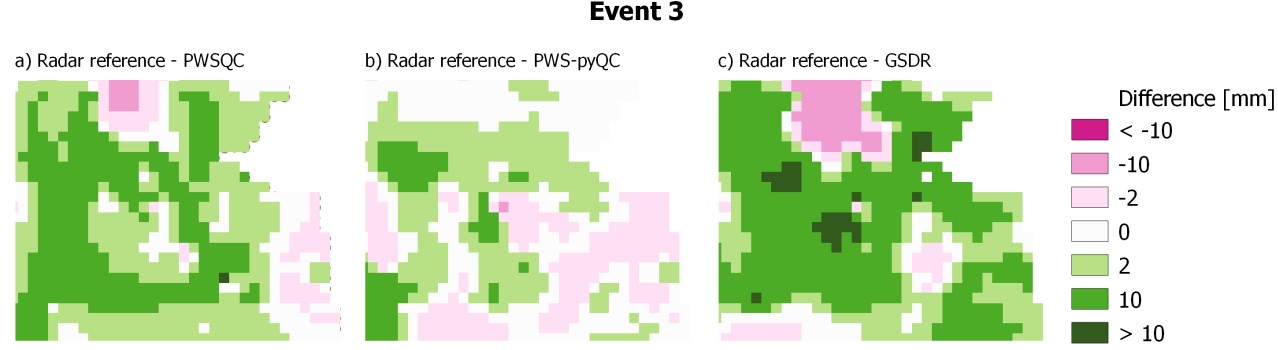

Difference [mm]
< -10
-10
-2
0
2
10
> 10

**Figure A6.** Differences between the radar reference and the interpolated maps from the three QC algorithms for event 3.

**Event 4**

a) Radar reference - PWSQC  b) Radar reference - PWS-pyQC  c) Radar reference - GSDR-QC

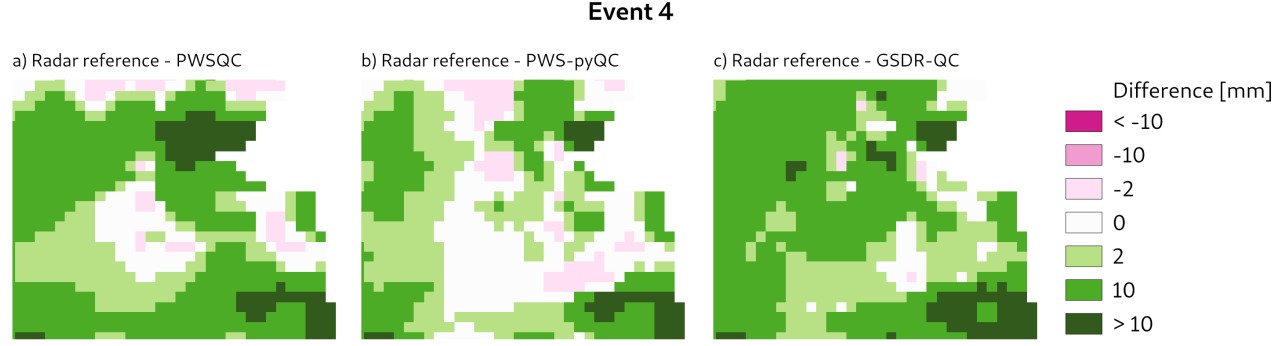

Difference [mm]
< -10
-10
-2
0
2
10
> 10

**Figure A7.** Differences between the radar reference and the interpolated maps from the three QC algorithms for event 4.

## Appendix B: Scatter plots for the four selected events

Figure B1 shows four scatter plots for the chosen events. The scatter plots are derived by comparing the hourly PWS data after QC has been applied, with a gauge-adjusted radar product, more specifically the overlying pixel of these PWS locations. Only the remaining hourly intervals for every QC method were considered. The data of PWSQC are displayed by the red dots, those of PWS-pyQC by the blue squares, and the GSDR-QC results by the green triangles. For every event, several metrics are calculated and showcased within each plot. For each QC method, the number of total data points in the event (134 PWS * 24 hours) that is covered after filtering is provided as a percentage. Given that we did not start off with 100% data availability in the original PWS dataset, this should only be interpreted relative to the other QC method outcomes. This shows that after PWS-pyQC, most data is rejected.

GSDR-QC shows more remaining data after QC, evident are the 0 mm precipitation records in PWS data, while the radar reference records rainfall (the dots spread out horizontally on the x-axis). This is due to faulty zero checks in the other two methods being implemented at the sub-daily timescale, whereas the GSDR-QC applies the check to daily aggregated data, resulting in reduced sensitivity to missing observations.

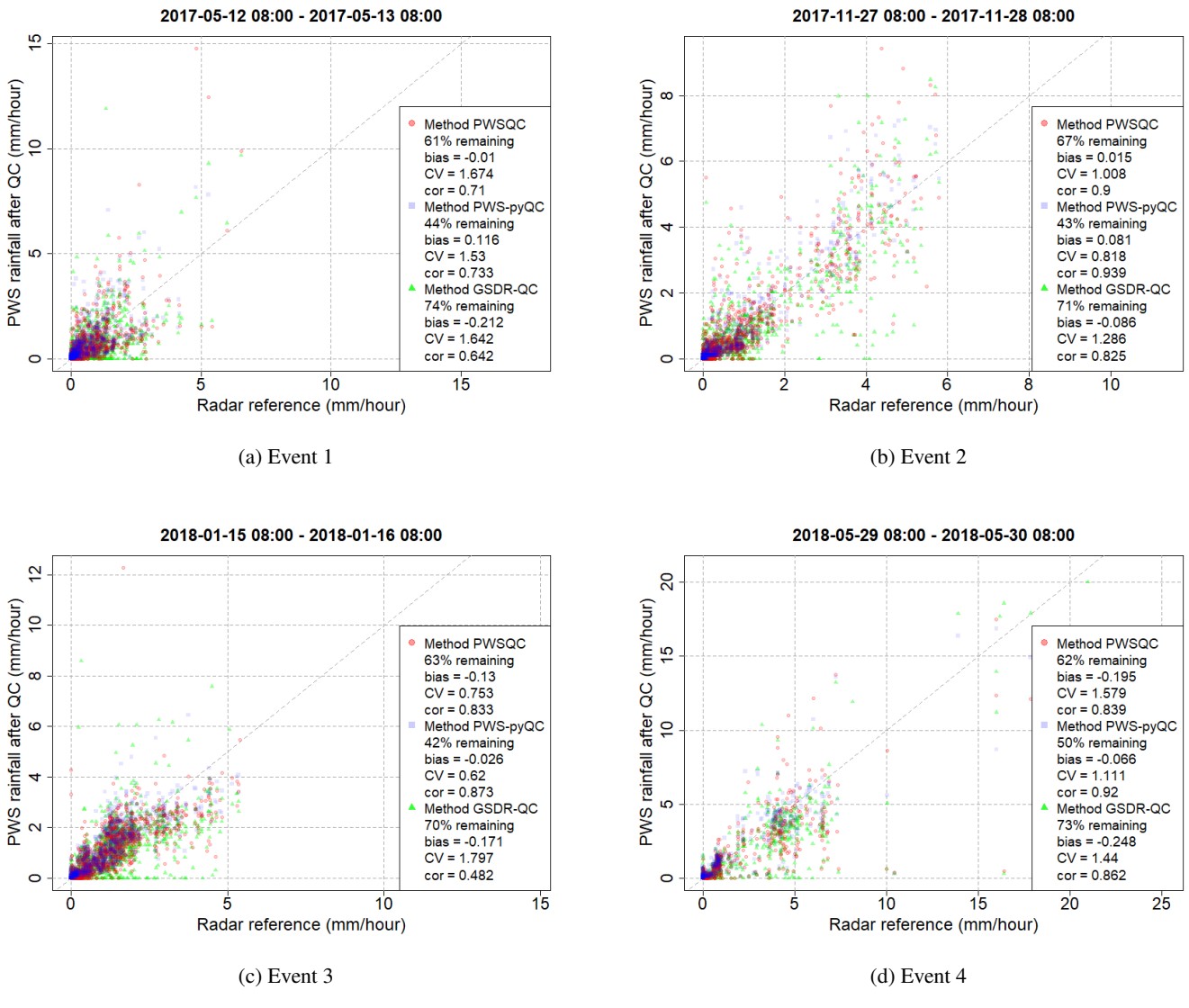

**Figure B1.** The scatterplots of hourly rainfall amounts of PWS after QC is applied against gauge-adjusted radar reference at the PWS location, including metrics for each of the three QC algorithms.

## Appendix C: Areal rainfall

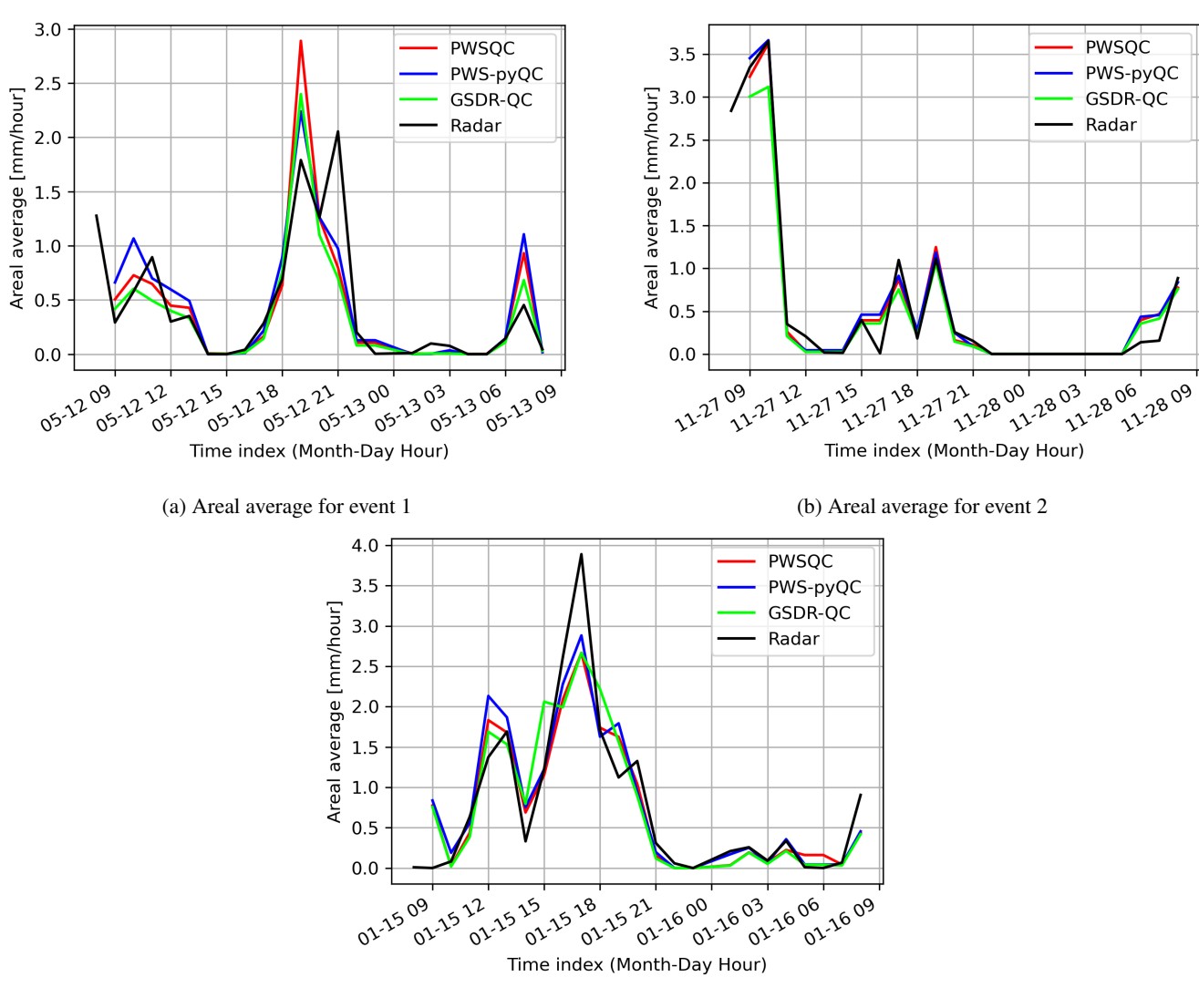

(a) Areal average for event 1

(b) Areal average for event 2

(c) Areal average for event 3

**Figure C1.** Panels (a), (b) and (c) show the areal rainfall over the Amsterdam metropolitan area for event 1, 2 and 3, respectively.

## Appendix D: Metrics calculated for the four events

**Table D1.** Comparison metrics calculated for the four events

|  |  | Event 1 | Event 2 | Event 3 | Event 4 |
|---|---|---|---|---|---|
| **Remaining PWS [%]** | PWSQC | 61 | 67 | 63 | 62 |
|  | PWS-pyQC | 44 | 43 | 42 | 50 |
|  | GSDR-QC | 74 | 71 | 70 | 73 |
| **Pearson Correlation** | PWSQC | 0.71 | 0.90 | 0.83 | 0.84 |
|  | PWS-pyQC | 0.73 | 0.94 | 0.87 | 0.92 |
|  | GSDR-QC | 0.64 | 0.83 | 0.48 | 0.86 |
| **Bias** | PWSQC | -0.01 | 0.02 | -0.13 | -0.20 |
|  | PWS-pyQC | 0.12 | 0.08 | -0.03 | -0.07 |
|  | GSDR-QC | -0.21 | -0.09 | -0.17 | -0.25 |
| **CV** | PWSQC | 1.67 | 1.01 | 0.75 | 1.58 |
|  | PWS-pyQC | 1.53 | 0.82 | 0.62 | 1.11 |
|  | GSDR-QC | 1.64 | 1.29 | 1.70 | 1.44 |