# Peer review of "Technical note: A guide to using three open-source quality control algorithms for rainfall data from personal weather stations"

_Hydrology and Earth System Sciences, 2023_

## Referee Comment (RC1)

Comments to the Author

Review of the paper "Technical note: Overview and comparison of three quality control algorithms for rainfall data from personal weather stations".

In this paper, the authors describe a new source of rainfall data: personal weather stations that promise to improve rainfall monitoring. However, their reliability may be seriously inferior compared to professional rain gauges, so quality control algorithms are necessary. Three quality control algorithms are applied to a PWS network based on four rainfall events in Amsterdam to demonstrate its performance compared to a radar gauge-adjusted KMNI product. The results show a better consistency in mean regional rainfall and an improvement in long-term correlation, although it does not perform well on rainfall maps. Finally, recommendations for the application of the three quality control algorithms are given.

The logic of this paper is clear and rigorous, the writing aspect is also good for reading. This author tries to give a comparison of the performance of PWS after different quality controls, but the limited amount of data (only 4 events) leads to the fact that the persuasive power is not strong, and, the analyses seem to be inadequate (also possible that due to the small number of samples being shown). There are also a few deficiencies in the writing of the paper, and this paper can be published after these are resolved. Therefore, my suggestion is major revision.

Below are some specific comments.
Major comments:
1. More information about rain gauges in PWS needs to be presented. Why does it exist? How does it differ from professional rain gauges? Is it just the operation being personal? What are the ways in which these private PWS data are contributed? What are their motivations?
2. Line 51: The gauge-adjusted radar product from the Royal Netherlands Meteorological Institute (KNMI) is used as a reference data set. Please describe its performance indicators.
3. Line 105: The aim of this paper is "a first demonstration of their applicability and performance". Would just four rainfall events be too little. Can the robustness of the assessment results be guaranteed? More rainfall events are expected.
4. "These rainfall events were selected in such a way that the majority of the PWSs registered significant rainfall for a large duration of time.". Whether PWSs do not exhibit significant rainfall in many rainfall events, even if significant rainfall does exist.
5. Figure 2: Both underestimated rainfall in the south-west of the region What is the reason? Is it because of the absence of stations here? It is helpful to have the stations labelled on the map to help understanding. Also it is expected that rain maps that have not been revised are placed.

Minor comments:

1. Line 9: Full name of the QC.

2. Line 47: "Figure 1" or "Fig. 1".

3. Figure 1: Please examine the chart carefully. Does the red rectangle in the small map in the upper left corner overlap with the red rectangle in the larger map? Also, the addition of longitude and latitude is helpful.

4. Line 67: "R software"?

5. Table 1: Text alignment in table.

6. eq. 3: '$x$' and '$y$' need to be specified.

7. Line 4: "May 2017 - May 2018", Line 47: "May 2016 and June 2018", whether or not it matches. Such a description is also misleading to the reader, as it seems to actually involve only four days.

---

## Author Comment (AC1)

**Response to reviewer 1**

We thank the reviewer for her/his comments. Please find our response in blue. We ask the reviewer to refer to the response to review #2 for more details on our course of action.

1. More information about rain gauges in PWS needs to be presented. Why does it exist? How does it differ from professional rain gauges? Is it just the operation being personal? What are the ways in which these private PWS data are contributed? What are their motivations?

   This will be added to the revised manuscript

2. Line 51: The gauge-adjusted radar product from the Royal Netherlands Meteorological Institute (KNMI) is used as a reference data set. Please describe its performance indicators.

   More information will be added on the gauge-adjusted radar product and our reason for choosing this in our technical note.

3. Line 105: The aim of this paper is "a first demonstration of their applicability and performance". Would just four rainfall events be too little. Can the robustness of the assessment results be guaranteed? More rainfall events are expected.

   We agree with the reviewer's comment that there is insufficient evidence to fully compare the efficacy of the algorithms based on the few selected events. The manuscript will be rewritten with more focus on the demonstration of how to use the QC algorithms, rather than on their performance (which was our original intention). Please see our response to review #2 for additional information.

4. "These rainfall events were selected in such a way that the majority of the PWSs registered significant rainfall for a large duration of time.". Whether PWSs do not exhibit significant rainfall in many rainfall events, even if significant rainfall does exist.

   This is a valid point in evaluating the performance of PWS in general (QC method not withstanding). However, in order to highlight the relative QC performance, we have to choose events where the majority of the raw PWS data had measurements of significant rainfall amounts, in order to showcase differences after QC has been applied. There have been other studies focusing more on PWS performance in general, see for example de Vos et al. (2017), where rainfall measurements of three PWSs are compared for a longer period next to a high-end rain gauge.

   **Reference paper:** [1] de Vos, L., Leijnse, H., Overeem, A., and Uijlenhoet, R.: The potential of urban rainfall monitoring with crowdsourced automatic weather stations in Amsterdam, Hydrol. Earth Syst. Sci., 21, 765–777, https://doi.org/10.5194/hess-21-765-2017, 2017.

5. Figure 2: Both underestimated rainfall in the south-west of the region What is the reason? Is it because of the absence of stations here? It is helpful to have the stations labelled on the

map to help understanding. Also it is expected that rain maps that have not been revised are placed.

*We appreciate the point raised and have investigated the location. The most likely reason for this is that the Amsterdam airport is located in this region and no PWSs are available. The station density will be discussed in the revised manuscript.*

**Minor comments:**

*We will account for the minor comments in the revised manuscript.*

1. Line 9: Full name of the QC.
2. Line 47: "Figure 1" or "Fig. 1".
3. Figure 1: Please examine the chart carefully. Does the red rectangle in the small map in the upper left corner overlap with the red rectangle in the larger map? Also, the addition of longitude and latitude is helpful.
4. Line 67: "R software"?
5. Table 1: Text alignment in table.
6. eq. 3: 'x' and 'y' need to be specified.
7. Line 4: "May 2017 - May 2018", Line 47: "May 2016 and June 2018", whether or not it matches. Such a description is also misleading to the reader, as it seems to actually involve only four days

---

## Author Comment (AC2)

**Response to reviewer #2**

We appreciate the reviewers' time and effort in reviewing our manuscript, which is evident from the level of detail and comprehensiveness of their reviews. Our aim with this paper has been to provide guidance to the reader on the nuanced differences in application of three existing open-source quality control algorithms, with an example case where they have been applied to the same dataset as case study. Our main take-away from the two reviews we have received has been that the paper in its current form comes across too much as an analytic benchmarking study instead. Most comments of the reviewers, while highly useful to improve such a benchmarking study, will steer away from this initial aim of the paper. It is our intention (as stated in the Outlook) to provide a thorough benchmarking analysis in a future manuscript in which the three algorithms will be compared and a conclusion regarding their performance will be provided. We do plan to benefit from the reviewers' comments for that future purpose. However, we do believe it best to restructure the current manuscript such that it better fulfils its intended aims, and also to keep within the page-restrictions of a technical note.

Our approach to do so is as follows:

1. We provide additional details about the three algorithms, e.g. how they work, the ability for real-time application, data requirements, and coding environment.

2. We will elaborate further on Table 1 (seen as a technical comparison of QC algorithms) and provide an additional table with operation guidelines.

3. Re-title the manuscript to "Technical note: A guide to using three open-source quality control algorithms for rainfall data from personal weather stations".

4. An aspect that was not clearly communicated is the availability of the three open-source algorithms in the OPENSENSE Sandbox. This allows users to freely (and easily) try the algorithms and grasp a feeling of how they perform and could be adapted for their 'personal' needs. The revised manuscript will also inform the reader how to operate the algorithms on the sandbox, including print-screens.

Please find our response in blue to some of the issues raised.

1. For each map of interpolated rainfall depths (such as Fig. 2), please use markers (small, maybe points) to indicate the PWS stations (using a legend to distinguish between stations used and discarded by each QC algorithm) and include (on a separate sub-plot) the interpolation of the raw data not passed through QC. This would help visualize how the tested QC algorithms change the raw data and the impact of the stations that were not filtered out.

   We have taken on board the point made, however as the QC algorithms work in different ways, the number of accepted/flagged PWS changes for every hour. In the currently shown figures, the 24-h rainfall depth is a summation of hourly interpolated maps with a changing number of PWSs for each hour as opposed to daily summations that are then interpolated. This means that the number of PWS contributions may differ for each hour and cannot be conveyed simply on a single map. For that reason, we chose to visualize in the figures the min, max, and mean number of used PWSs over

the 24-hour time window. This will be better explained in the revised manuscript. However, it is possible to include maps showing the interpolation using the raw data (including station markers) to the plots.

2. Please include latitude and longitude in all maps. Use the same grid both in Fig. 1 (case study area with stations) and all maps of interpolated precipitation depths, to facilitate any comparison. In the discussion of the results, you consider the radar rainfall product (referred to as radar reference) as aprioristically correct, since the interpolated, quality-controlled results from the PWS network are directly compared to the former (Fig. 2 and 3), and the performance of the QC algorithms is measured in terms of how closely the derived rainfall depth maps match the radar product. However, I do not think that this is the best approach, conceptually. While data from PWSs may contain errors because of the different reasons listed in your Introduction, the dramatically higher spatial resolution of the PWS network, compared to the official station network (KNMI) used for the correction of the radar rainfall product (134 vs. 1 station, in your case study area), implies that the information obtained from the PWS network may result in a potentially more detailed representation of the spatial distribution of precipitation depths, compared to the radar rainfall product, at least in some locations and time steps. While there may be errors in the PWS time series due to the potentially non-optimal usage of some stations by non-professionals, the radar product is also likely affected by other types of errors and uncertainties, inherent to the procedures for deriving that product. As such, these two sources of rainfall information should not be used as if one were an axiomatic truth and the other had to closely match the former, but rather as complementary sources, where each one has its own advantages and limitations.

We fully agree with the reviewer. The radar product that is used is a climatological product with radar rainfall depth corrected with information from two rain gauge networks, that is constructed with a considerable delay (i.e. not real-time available). This data has also been used as ground truth in the reference works [1] and [2] (please see list of references). As for this period there is no other reference with better quality to use for spatially distributed rainfall fields we opted for this radar product. The benefit of PWS is its availability in real-time and can be used as complementary information. The data from PWS and the weather radar should indeed be complementary and if possible merged to gain the most information (reference work [3] provides such a workflow). We feel that using this radar product is a best effort to validate the spatial-temporal dynamics of the interpolated rainfall fields, the limitations of using it as ground-truth will be noted more strongly in the revised manuscript.

**Reference papers:**
[1] de Vos, L. W., Leijnse, H., Overeem, A., & Uijlenhoet, R. (2019). Quality control for crowd-sourced personal weather stations to enable operational rainfall monitoring. Geophysical Research Letters, 46, 8820–8829. https://doi.org/10.1029/ 2019GL083731

[2] de Vos, L. W., A. Overeem, H. Leijnse, and R. Uijlenhoet, 2019: Rainfall Estimation Accuracy of a Nationwide Instantaneously Sampling Commercial Microwave Link Network: Error Dependency on Known Characteristics. J. Atmos. Oceanic Technol., 36, 1267–1283, https://doi.org/10.1175/JTECH-D-18-0197.1.

[3] Overeem, A., Leijnse, H., van der Schrier, G., van den Besselaar, E., Garcia-Marti, I., and de Vos, L. W.: Merging with crowdsourced rain gauge data improves pan-European radar precipitation estimates, Hydrol. Earth Syst. Sci. Discuss. [preprint], https://doi.org/10.5194/hess-2023-122, in review, 2023.

3. When considering the radar rainfall product (referred to as radar reference) aprioristically correct (see previous comment), you risk introducing some bias in the comparison of QC algorithms, as one of them (PWS-pyQC) expressly requires a reference observation network (primary network) to perform most of the bias correction procedures, and your provided reference observation network is a random subsample (20 pixels) of the radar rainfall product itself. As the output from PWS-pyQC is then more likely to closely match the radar reference by construction (as it happens in Fig. 2), you might end up always considering PWS-pyQC as superior over the other QC algorithms in obtaining values of precipitation depth close to truth. The next point suggests performing a sensitivity analysis that might also help assess if there is indeed such a bias, and to what extent. Show on a map the random 20 pixels of the radar rainfall product used as the primary network for PWS-pyQC (line 83). Perform a sensitivity analysis trying with other random subsamples, also playing with the number of pixels considered. This may help shed more light on the sensitivity of the performance of PWS-pyQC to the size and configuration of the primary network. Specifically, the stability of the performance of PWS-pyQC for a range of different subsamples of fixed size (i.e., fixed number of pixels) will be an indicator of the robustness of the methodology, while any observed changes in the average performance with the number of pixels in the primary network will help get an idea of how much information from the primary network is required in general for applications on other case study areas.

   We fully agree with the reviewer that such a sensitivity analysis would be beneficial for a benchmarking study but we consider this is beyond the scope of this technical note, which is intended as a guide to the usage of the QC algorithms.

4. PWSQC and GSDR-QC both include some procedures that are based on neighbor-checks (lines 58, 61, and 95; Tab. 1). E.g., PWSQC detects faulty zeros at those stations with 0 mm observations but concurrent precipitation at some neighboring stations (line 58); PWSQC also detects outliers based on the correlations of concurrent records at neighboring stations (line 61). I was left wondering what the criteria are to identify neighboring stations, e.g., what is the maximum distance beyond which two stations would not be considered neighbors of each other? I would assume that there are some control parameters that the modeler can set when using the algorithms. Have you tried playing with them? It would be interesting to see a sensitivity analysis to these and any other control parameters that may be available to the modeler (if applicable). This would ensure that the comparison of the three QC algorithms is made when the optimal parameter settings are adopted for each of the tested approaches.

   Additional information regarding this aspect will be added. The sensitivity analysis is also important but is beyond the scope of this technical note.

5. Figure 2: Both underestimated rainfall in the south-west of the region What is the reason? Is it

because of the absence of stations here? It is helpful to have the stations labelled on the map to help understanding. Also it is expected that rain maps that have not been revised are placed.

*In that location is the Amsterdam airport and no PWSs are present. As the number of used PWSs changes for every hour and every QC, it is rather difficult to visualize them for a daily event. Please see our response to point 1.*

6. Table 1 effectively summarizes the characteristics of the three QC algorithms. However, I feel that some of the concepts mentioned there (e.g., "Level of QC-allocation", "WMO QC classification types") also need to be briefly introduced in the body of the manuscript. I would also describe in a little more detail how the routines in each QC method work, e.g., are the neighbor-checks iterative processes? In general, I feel that a deeper understanding of alternative algorithms is important to guide informed decisions on which is the most suitable algorithm for the specific case at hand.

*We will elaborate further on Table 1 (seen as a technical comparison of QC algorithms) and provide an additional table with operation guidelines.*

7. One of the two main findings of the work involves the issue with the presence of faulty zeros, and in particular the lower performance of the GSDR-QC algorithm in detecting faulty zero observations, compared to the other two algorithms, as remarked, e.g., while commenting on Fig. 2 (line 143), or in the Discussion (line 180) and Conclusions (line 200) sections. These conclusions are based on the observation of "dry spots" (Fig. 2) in the interpolation maps obtained from the quality-controlled PWS data, as compared to the radar reference, which are most noticeable for GSDR-QC. However, it seems that many of those spots occur at locations with limited station availability, compared to the rest of the PWS network in the study area. E.g., focusing on Fig. 2b, c, and d, the dry spots are located in a big portion of the SW (south-west) quadrant and two smaller portions in the NW and SE quadrants of the case study area, where there are noticeable gaps in the station network (see Fig. 1). Hence, the lack of sufficient data may be an alternative explanation for the presence of dry spots, as the lack of stations may cause problems with the QC algorithms (e.g., with the neighbor-checks). In turn, this means that the observations of dry spots may not be connected a priori to the presence of any large number of faulty zeros in the original data (in this respect, the suggestion given in point 1 would be very helpful to give the right interpretation), but rather be a consequence of the specific experimental settings. The lack of a univocal interpretation of why dry spots are observed in the maps of interpolated precipitation depths prevents an objective assessment of the performance of the QC algorithms. What you can say at this stage is that PWS-QC and PWS-pyQC are apparently more effective than GSDR-QC in avoiding underestimation of precipitation depths at locations with limited presence of stations. I would be confident with concluding that GSDR-QC has a lower sensitivity to faulty zero observations only if those dry spots were observed in portions of the study area with plenty of stations, and where the raw data actually present some faulty zeros (but the raw data are not shown in this current version of the manuscript), so that to avoid any external source of bias and only focus on the intrinsic limitations of the algorithms (if any). It may be a good idea to consider more than one case study area and more events per area, to derive more robust, less biased conclusions about the performance of each QC algorithm. When selecting a larger number of events, I would suggest not to limit the search to those with significant rainfall all over the study area

and for a long duration (as you state in line 108), as it may be more useful to test the QC algorithms under a wide range of heterogenous conditions.

The presence of faulty zeros was noted when comparing the interpolated data to the reference radar values at the same pixel locations (please see the scatter plots for the four selected events). Without trying to interpret too much from the limited case study, we will add information regarding possible reasons why GSDR-QC fails to identify false zeros. We agree however with the raised point and we will rephrase the text and the conclusion.

8. The other main finding of the work involves the tendency to underestimate the peak in precipitation depth in the maps of interpolated values after performing the QC, as highlighted in the Discussion (line 175) and Conclusion (line 199) sections. However, also the robustness of this conclusion is, in my opinion, undermined by the presence of spots of low PWS network density in the case study area, as in many cases (e.g., Fig. 2, A1, A2) the underestimation of the precipitation depth occurs at locations affected by the lack of stations, as you also admit in line 176. Because of the specific conditions of the case study considered, the underestimation of the peak may be imputable to the lack of a sufficient number of stations in the location of the peak, and not to limits in the way the QC algorithms are designed (at least in principle). This article is about comparing three QC algorithms, but how can you assess pros and cons of these if the precipitation event that you consider (event 4) displays its peak in a location where it seems there is no sufficient data coverage in the first place? To draw robust conclusions about how these algorithms work and compare their performances, you should try to have almost ideal conditions in terms of data coverage, so as to focus solely on any problems with the algorithms.

The aim of the QC algorithms is to make the best use of the opportunistic data that is available. The underestimation of the rainfall peaks can be related to the absence of stations at the location of the maxima and is not related to the QC algorithm itself. The effect of the network density on the observed rainfall maxima was described for example by Bárdossy et al. (2023). We will rephrase the manuscript to avoid drawing conclusions regarding the underestimation and its relation to the QC algorithm.

**Reference paper:** [4] Bárdossy, A. and Anwar, F.: Why do our rainfall–runoff models keep underestimating the peak flows?, Hydrol. Earth Syst. Sci., 27, 1987–2000, https://doi.org/10.5194/hess-27-1987-2023, 2023.

9. How did you choose the rectangle for the case study? E.g., it seems that the distance between the edges of the study area and the most peripheral PWSs of the network is not the same on each side. Please describe how the boundary of the study area was outlined and provide the coordinates of its corners. It is worth noting that the choice of the rectangle may have some effects on the computation of the areal averages, and in turn on the plots in Figures 4 and C1.

In this case study an open access published dataset (https://doi.org/10.1029/2019GL083731) was used that was the basis for the analyses in de Vos et al. (2019). The data set consists of all PWSs with a rain module in the Amsterdam metropolitan area, defined as the area between 4.67–5.05 degrees longitude and 52.24–52.44 degrees latitude ($\approx 575\ km^2$) between 1 May 2016 and 1 June 2018. This is to our knowledge one of the few open access Netatmo PWS dataset. The reviewer

raises a valid point regarding the edge effect, which means indeed that the boundaries of the box will have a lower accuracy than the center of the study area. This will be added as a remark in the revised manuscript.

**Reference paper:** [1] de Vos, L. W., Leijnse, H., Overeem, A., & Uijlenhoet, R. (2019). Quality control for crowdsourced personal weather stations to enable operational rainfall monitoring. Geophysical Research Letters, 46, 8820–8829. https://doi.org/10.1029/ 2019GL083731

10. Why did you choose 15 validation locations from the radar reference (line 164)? For each of these locations, did you obtain one Pearson correlation metric by Eq. (1), and do the boxplots in Fig. 5 show the distribution of the values of this metric across the different validation locations? Or is it the other way around, i.e., the Pearson coefficient is computed across the different validation locations, and the boxplot shows the variability of the Pearson coefficient with time? Please clarify this in the manuscript.

   The 15 locations were chosen as they provided more or less homogeneously distributed locations within the study area. The time series at these locations was estimated using the 3 QC outputs and compared to the observation time series. This provided 15 Pearson correlation values across the validation locations, which are summarized in the boxplots.

11. I personally find it a bit unclear how the presented results lead to some of the concluding remarks in the Conclusion. E.g., line 206: "the PWSQC algorithm is most useful where there is a dense PWS network, and the GSDR-QC is most appropriate in locations where the PWS network is sparse and comprises rain gauges from a range of manufacturers (resulting in a range of potential errors)". Please elaborate more on that in the article.

   We agree that the limited number of case studies does not allow far-reaching conclusions. We will restructure the manuscript to use the selected events as a case study. These are indicative but not conclusive. Please refer to the introductory paragraph on our course of action.

   **Minor comments:**
   We will account for the minor comments in the revised manuscript.

   In Eq. (2) and Eq. (3), x and y without any hat or subscript are used but not defined.
   The CV in Eq. (3) is different from the typical definition for a single variable, as in the numerator there is the standard deviation of the difference between x and y (you use sigma as the standard deviation operator, correct?), while you consider the average y in the denominator. Please provide some more information.
   Line 149: "GSDR-QC shows the most remaining data after QC...". I would rephrase it as "GSDR-QC retains more PWS stations, as compared to ...", or something like that.

---

## Author Response (AR1)

Dear Editor,

we tried to include our responses to the reviewers as best as we could. Please check the updated manuscript and the one with the track changes.

Thank you,
Abbas El Hachem

---

## Author Response (AR2)

We thank the reviewers and the Editor for their positive Response. We accepted and included all the minor corrections.